# Direct Shoot Regeneration from the Finger Millet’s In Vitro-Derived Shoot Apex and Genetic Fidelity Study with ISSR Markers

**DOI:** 10.3390/biotech14020029

**Published:** 2025-04-18

**Authors:** Theivanayagam Maharajan, Veeramuthu Duraipandiyan, Thumadath Palayullaparambil Ajeesh Krishna

**Affiliations:** 1Division of Plant Molecular Biology, Entomology Research Institute, Loyola Collège, Chennai 600034, India; susirajan143@gmail.com (T.M.); ajeeshkrishnatp@gmail.com (T.P.A.K.); 2Division of Plant Molecular Biology and Biotechnology, Department of Biosciences, Rajagiri Colle of Social Sciences, Kochi 683104, India

**Keywords:** clonal fidelity, finger millet, tissue culture

## Abstract

Globally, people are cultivating finger millet, an important cereal, to improve food availability and health benefits for humans. However, the biotechnological research on this millet is limited and insufficient in this field. The primary focus of this study is to optimize an efficient regenerated protocol for initiating further plant transformation studies, using the shoot apex as an explant and various growth regulators. For example, three cytokinins (BAP, TDZ, and Kin) at different concentrations were used to induce multiple shoots of finger millet. Among these, TDZ (4.5 µM) provided the maximum number (17.3) of shoots as compared to BAP and Kin. IBA (2.46 µM), along with MS medium, was used for the induction of roots, where 5.6 roots were produced in an individual shoot and the length of the root was longer with a size of 8.2 cm after two weeks of incubation. The clonal fidelity of the in vitro regenerated plantlets of finger millet was confirmed by ISSR primers. Overall, the present work developed a robust and reliable procedure for the establishment of efficient and reproducible regeneration through the shoot apex that will be useful for the genetic improvement of this crop. The genetic enhancement of these millets as well as the successful creation of transgenic plant varieties modified for resistance to biotic and abiotic challenges in the near future would be aided by this study.

## 1. Introduction

Millets are a group of small-seeded cereals, classified into major and minor millets based on the size of the grains [1]. They are the most secure crops for small farmers since they are the hardiest, most resilient, and most climate-adaptable crops in tough, hot, and drought scenarios [2]. More than seven millets (including finger millet) are mostly cultivated worldwide. Finger millet is an important primary food especially for rural populations of southern India and East and Central Africa [3]. It can be grown under a wide range of situations, i.e., from sea level to hilly regions, but a very good yield is obtained under well-drained loamy soil. This magical millet contains a higher amount of calcium than other cereals and milk [4]. Since its fat contains zero cholesterol and many macro- and micro-nutrients, its consumption instead of milk is becoming an option for vegetarians [5]. Finger millet is nutritionally comparable to major cereals and serves as a good source of protein, dietary fiber, and more [6]. However, biotic and abiotic variables drastically affect the yield and productivity of finger millet. Therefore, there is an urgent need for innovative strategies and technology to boost finger millet production globally. It has been believed that in vitro research is crucial for improving finger millet [7]. In vitro research is a method that involves the cultivation of new platelets by utilizing plant material in a growing medium. The initial plant material is cultivated and developed in a highly controlled and specific environment. The plantlets are obtained in a brief period of time using a minimal amount of plant tissue. The novel plants that have been produced are free of any diseases. The plants can be cultivated year-round, regardless of the season. The tissue culture technique does not require a substantial amount of space to cultivate the plants. Few researchers have been making efforts to optimize tissue culture protocols for a few cultivars of finger millet. In a previous study, different types of finger millet explants have been successfully used for plant regeneration [8]. Until the present date, the shoot regeneration of finger millet was developed through indirect organogenesis using different types of explants and methods. However, the direct regeneration method was not much emphasized. Some millet researchers have attempted to develop finger millet through direct organogenesis methods. For example, Satish et al. (2015) optimized direct regeneration protocol for various genotypes of finger millet using different plant growth regulators [9]. Likewise, various genotypes of finger millet were also used to optimize the direct regeneration protocol of finger millet [8].

Direct plant organogenesis is an effective method to decrease somaclonal variation significantly, as it minimizes the culture duration, callus formation, and sub-culturing cycles [10]. By comparing the plantlets in our experiment to the mother plant using inter-simple sequence repeats (ISSR) marker analysis, no somaclonal change was observed. Using ISSR markers, in vitro regenerated plants were found to have somaclonal variation in a prior work [11,12,13]. Compared to regeneration by embryogenesis, finger millet direct organogenic systems have been the subject of far less research. As of the present, there are just two reports on effective plant regeneration in finger millet genotypes using direct organogenesis. Therefore, it is imperative to implement the direct organogenesis procedure in order to enhance the variety of finger millet that is best suited to India’s tropical and subtropical climate. The aim of the study is to optimize an efficient and robust tissue culture protocol for finger millet, using various growth regulators and the shoot apex as an explant to initiate further transformation work on this millet. In this work, we have attempted to use thidiazuron (TDZ) as a plant growth regulator for the finger millet primary shoot apex. In the near future, this research will be beneficial for the successful production of transgenic plant varieties resistant to biotic and abiotic challenges as well as for the genetic enhancement of millets.

## 2. Materials and Methods

### 2.1. Plant Material and Explant Preparation

The International Crops Research Institute for the Semi-Arid Tropics (ICRISAT), located in Patancheru and Hyderabad, India, provided the finger millet variety IE-2606. After 15 min of rinsing under running water, the seeds were treated for 5 min with 2–3 drops of 0.02 percent tween-20 solution, and then they were cleaned with water to remove any sticky dust. After surface-sterilizing the seeds for five minutes with 0.1 percent HgCl2, they were surface-disinfested for thirty seconds with 70% ethanol. After that, the seeds were steeped for six hours and cleaned three times in sterile distilled water. For seed germination, we used Murashige and Skoog (MS) basal media (Bacteriological grade; HI Media, Mumbai, India). The pH was adjusted to 5.7 before autoclaving at 121 °C for 15 min. The MS basal medium (25 mL) was poured in Petri dishes (90 £ 15 mm) and 10–15 soaked seeds were placed in each Petri dish. Parafilm was used to seal the Petri dishes, which were kept at 25 ± 2 °C with a dark photoperiod and 78 percent relative humidity (RH). For four days, the cultures were incubated in the dark. The shoot apex was removed from the seedling after it had been incubated for four days, and it was employed as the explant source for multiple shoot induction in the studies that followed.

### 2.2. Shoot Multiplication and Elongation

For multiple shoot induction, the immature shoot apex was transferred to MS basal media associated with three plant growth regulators such as TDZ, N6-benzyl amino purine (BAP), and BAP with kinetin (Kin). Glass bottles were used as a culture vessel to maintain the plants and about 30–50 mL of culture was maintained in each bottle to maintain the plant culture. Initially, about 3 explants at each concentration were maintained in each bottle for multiple shoot induction. After multiple shoots were initiated, they were maintained in a single bottle for multiple shoot induction. The cultures were incubated for 12 days at 26 ± 2 °C in a growth chamber with 16/8 h light/dark and 75/50 percent RH. The cultures were checked for morphological changes every two days. For the purpose of promoting shoot proliferation, an ideal concentration of the phyto-hormone combination causing profuse multiple shoot induction was chosen. In order to assess the effects of the multiplication media on shoot multiplication and elongation, shoot clumps were further sub-cultured once more in the identical TDZ, BAP, and BAP with Kin shoot induction medium. After five to six weeks of culture, the proportion of elongated shoots that formed, the mean length of the proliferating shoots, and the mean number of shoots per explant were determined.

### 2.3. Root Induction

After being aseptically divided, individual shoots were placed in MS basal medium supplemented with varying amounts of Indole-3-acetic acid (IAA) (0, 0.25, 2.85, and 5.70 µM) and Indole-3-butyric acid (IBA) (0, 0.25, 2.46, and 4.9 µM) to induce root growth. For the induction of roots, plants were maintained in 100 mL test tubes. A single plant was maintained in each tube. The cultures were grown at 26 ± 2 °C for 16/8 h light/dark for 2 weeks. The regenerated shoots’ roots were permitted to grow to a length of roughly 5 cm in vitro. After two weeks of incubation, the proportion of shoots with roots, the mean number of roots, and the average length of the roots were noted.

### 2.4. Acclimatization

After being removed from the culture vessels, the shoots with completely formed roots were thoroughly cleaned to remove the extra agar using sterile distilled water. The plantlets were then placed in single-use cups with sterile soil and vermicompost (1:1 *w*/*w* ratio), and, to retain humidity and allow air circulation, they were covered with perforated plastic bags. For three weeks, the transplanted plantlets were left to acclimate to a temperature of 25 ± 2 °C and a light/dark cycle of 16 h per day. After the second week, the plastic bags were taken out of use. Following a three-week period of acclimation, the plantlets were moved to separate pots filled with a variety of soil (soil/sand/vermicompost in a 2:1:1 ratio). After being moved inside the greenhouse, the plantlets were given regular irrigations to keep the soil’s water level between 75 and 80 percent of its total capacity.

### 2.5. Analysis of ISSR Primers

ISSR markers were employed to verify the genetic integrity of the plants that were regenerated in vitro. Five (randomly) of the seventeen regenerated finger millet genotype (IE-2606) plants were examined using five ISSR markers in comparison to the mother plants. Using the CTAB technique, genomic DNA was extracted from the leaves of both the mother plants and regenerated plants. The PCR reactions were carried out using previously published procedures [14]. After being separated in 2% (*w*/*v*) agarose gels at 80 V, the amplification products were stained with ethidium bromide. By comparing the PCR results’ fragment sizes to 100 base pair (bp) DNA ladders, their sizes were approximated.

### 2.6. Statistical Analysis

One-way ANOVA was used to find changes between means that were statistically significant (SE). SPSS 17.0 (SPSS Inc., Chicago, IL, USA) was used to analyze the data, and values were thought to be significant if *p* < 0.05.

## 3. Results

### 3.1. Preliminary Study on Shoot Induction by Cytokinin

After four days of inoculation on plant growth regulator-free MS media, mature finger millet seeds that had been surface-sterilized began to form shoots. In the present study, we have tested concentrations for three cytokinins including TDZ, BAP, and combinations of BAP and Kin (Table 1). With all TDZ concentrations present in MS medium, the best response to shoot induction was seen; both BAP alone and BAP with Kin produced the best response at 8.8 µM (BAP) and 8.8 + 3.4 µM BAP + Kin (Table 1). We have observed that TDZ produced the higher number of shoots per shoot apex compared to the BAP and Kin combination. In this study, the detected concentrations were utilized for follow-up research.

### 3.2. Multiple Shoot Induction

This genotype reacted favorably to every TDZ concentration and PGR combination examined in the shoot induction medium. The maximum number of shoot inductions (17.3) was observed in 4.5 µM of TDZ followed by 2.2 µM (10.0), 6.8 µM (15.0), and 9.0 µM (13.0) of TDZ. Among various concentrations, BAP with Kin (8.8 + 3.4 µM) produced the highest number of shoots (14.6) and 8.8 + 1.1µM BAP with Kin produced the lowest number of shoots (8.3) (Table 1). BAP alone induced the maximum number of shoots (9.3) in 8.8 µM followed by 2.2 µM (3.6), 4.4 µM (6.3), and 6.6 µM (8.0). It has been seen that the combination of BAP and Kin produced fewer shoots, whereas TDZ produced a considerably higher number of shoots, which were also longer (Figure 1B–D). Based on the concentration and mix of PGRs in the induction medium, there were variations in both the quantity of shoots per explant and the response to shoot induction. The number of induced shoots at the shoot apex treated with TDZ increased in the following weeks, and up to 15–17 shoots grew from the shoot apex explant on 4.5 µM of TDZ.

### 3.3. Shoot Elongation and Maturation

The MS medium with 4.5 µM of TDZ showed the best response in terms of shoot multiplication. Hence, the same concentration was used for the shoot elongation process of this study (Figure 1E). The maximum average length of 7.4 cm was observed in TDZ (4.5 µM). Therefore, the other three concentrations of TDZ produced 2.2 µM (3.1 cm), 6.8 µM (6.3 cm), and 9.0 µM (6. 0 cm), respectively. BAP 8.8 µM produced the highest shoot length (5.9 cm) compared to 2.2 µM (1.6), 4.4 µM (4.1 cm), and 6.6 µM (5.2 cm). Shoot proliferation and elongation were obtained from 4.5 µM of TDZ and no further sub-culture was performed.

### 3.4. Induction of Roots

Using different IAA and IBA concentrations, rooting was induced. After being removed, the sub-cultured shoots were placed in half-strength MS media that contained 0.0 µM, 0.25 µM, 2.85 µM, and 5.70 µM IAA and 0.0 µM, 0.25 µM, 2.46 µM, and 4.9 µM IBA. After 10–14 d, the roots were well developed from the plantlets. Compared to IAA, IBA added to the MS medium resulted in a better rooting response (Figure 1F). The IBA (2.46 µM) significantly increased the rate of rooting growth and the root length compared to other concentrations of IBA (Table 2). IAA (2.8 µM) added to the MS medium produced more roots (4.6) and the longest roots (6.9 cm) compared to other concentrations of IAA (Table 2). The poorest root induction and growth were observed in 0.0 µM in IAA and IBA, respectively.

### 3.5. Hardening and Acclimatization

Following an 8-week culture period, the plantlets generated roots during a 16 h photoperiod. The rooted shoots were transferred to plastic cups (Figure 1G) with a 100% survival rate, 40 days after transplanting. When compared to in vivo plants established from seeds, all in vitro regenerated plants grew well and showed no variation in phenotypic uniformity, including growth appearances, maturity, flowering, seed setting, and grain yield (Figure 1H).

### 3.6. Examining the Clonal Fidelity of In Vitro Regenerated Plants by ISSR Markers

In this work, the clonal fidelity of in vitro regenerated finger millet plantlets was analyzed using five ISSR primers (Table 3). Each ISSR primer had varying numbers of scoreable bands: five (UBC-834) to seven (UBC-841) (Figure 2A,B). Five ISSR primers yielded a total of twenty-seven unique and scoreable bands, with an average of 5.4 bands per primer. Each primer banding exhibited a distinct set of amplifications, ranging from 100 to 1150 bp. The banding profiles of the plantlets that were regenerated in vitro were all monomorphic and resembled those of the parent plant. Additionally, when comparing the in vitro regenerated plants to the mother plant, no polymorphism was found. It is now shown that there was no somaclonal variation caused during clonal propagation and that the in vitro regenerated plants were genetically identical to their mother plants.

## 4. Discussion

For efficient direct shoot induction and plant regeneration, the shoot apex was employed as the first explant in the current investigation. Compared to other explants, shoot apex explants from millets are easy to handle, readily available, homogenous, and rapidly regenerate many shoots, making them great resources for biotechnology applications [10,15,16]. No effective methodology is now available for the direct regeneration of any species of millet. Thus, significant efforts have been undertaken in direct organogenesis on finger millet in recent years [9]. But this attempt ended up producing only 6.2 shoots in CO (RA) 14 using 17.6 µM BA, which was the lowest number reported in the finger millet genotype. In our previous report, finger millet genotype GPU 45 developed 26 shoots from the shoot apex explant in 4.5 µM TDZ and 4.6 µM Kin using somatic embryos [15]. In the present study, 17 shoots were produced in 4.5 µM TDZ alone in genotype IE-2606 using the shoot apex. This is the first report with the highest number of multiple shoot inductions in finger millet genotypes through direct organogenesis. This is consistent with earlier research on numerous shoots formed from the shoot apex through direct organogenesis in pearl millet [17], rice [10,16], maize [18], barley [19], and oats [20]. Three distinct cytokinins, including TDZ, BAP, and Kin, were employed in the current investigation to induce numerous shoots. The most successful hormone was identified to be TDZ, which also produced the greatest amount of multiple shoots, whereas BAP and Kin produced the fewest number of shoots. Satish et al. (2015) reported that multiple shoots were produced using BAP, whereas TDZ and 2, 4-D produced the least number of shoots in different types of finger millet genotypes [9]. The TDZ (4.5 µM) concentration was similar in the present study and in the previous study, but the incubation time was varied [9]. In the previous study, the shoot apical meristem was incubated for 4 weeks to induce multiple shoots, whereas in this study, the repeated sub-culturing of the shoot apex for 6 weeks induced a greater number of multiple shoots. The explant that was further incubated for 2 weeks yielded the greatest number of shoots. This is the first study on the genotype of finger millet that yields the most shoots in 4.5 µM TDZ.

TDZ, a non-purine cytokinin compound, exhibits a stronger effect in shoot proliferation in crop plants than BAP on in vitro morphogenesis [21,22]. In general, TDZ is an active hormone in plant species compared to other cytokinins [23]. Hence, TDZ is one of the most important cytokinins for shoot proliferation in monocot crop plants. This is consistent with a prior study where the maximum number of shoots was produced by a medium enriched with TDZ in *A. judaica* [24], *Rauvolfia tetraphylla* [25], *Artemisia vulgaris* [22], *Vaccinium vitis-idaea* [22], *Carthamus tinctorius* [26], *Cajanus cajan* [27], and sugarcane [28]. In barley, wheat, oats, and *Triticosecale*, multiple shoots were induced from mature embryos in 4.5 µM and 9. 1 µM TDZ [29]. Sujatha and Kumari (2007) reported that a 4.54 μM concentration of TDZ is required for efficient direct regeneration in *Artemisia vulgaris* [22]. TDZ (4.5 μM) also produced multiple shoots in different types of explants such as the mature embryos and apical meristems of oats [30]. Guo et al. (2011) also reviewed that low amounts of TDZ promote more shoot multiplication compared with other cytokinins [31].

Lata et al. (2013) suggested that the improvement of efficient micro-propagation in plants is an important aspect for the development in biotechnology applications [32]. Our current study aims to provide an effective technique for the genotype mass propagation of finger millet. Additionally, on TDZ-enriched media, shoots were regenerated from the finger millet shoot apex. This is the first report that we are aware of regarding the direct induction of shoots and subsequent plant regeneration using the shoot apex of the finger millet genotype IE–2606. The direct plant regeneration method is very useful for the production of many plantlets with 100% homogeneity conditions and also minimizes somaclonal variation, reducing the culture duration and sub-culturing cycles [9,10]. The direct plant regeneration method has been commonly used in many plants species such as *Cuminum cyminum* [33], *Alstroemeria* [34], *Vitis vinifera* [35], *Morus* [36], *Embelia ribes* [37], *Pistacia vera* [38], *Plumbago* [39], *Vitex negundo* [40], *Gossypium* [41], *Pigeon pea* [42], *Psidium gujava* [43], and finger millet [9].

In this experiment, IAA and IBA played a significant role in root regeneration. After two weeks of incubation in half-strength MS media containing IAA and IBA, roots spontaneously formed. The greatest quantity and longest roots were formed in the MS media with the addition of IAA and IBA. In terms of root induction, more roots were formed in the MS medium containing 2.4 µM IBA than in that containing IAA. This is consistent with a prior study on barley, which indicated that effective root induction required a full-strength MS medium with 2.4 µM IBA [44]. It has been found that many plant species, including *Morus*, root more readily in MS medium containing IBA [45], and this is also demonstrated for *Saccharum officinarum* [28] and *Rauvolfia tetraphylla* [25].

When comparing the in vitro regenerated plants to the parent plant, the ISSR data revealed no polymorphic bands. The absence of somaclonal variation among the regenerated plantlets was corroborated by these findings. Several molecular markers, including ISSR, have been utilized in the field of plant research to assess clonal fidelity in plants grown in tissue culture. Compared to other DNA markers, ISSR markers are more suited to identify differences across plants grown in tissue culture. This is because other markers are quite expensive and need a significant investment in both labor and equipment. Due to their versatility, affordability, and ease of use, ISSR markers were selected for this investigation in place of others. The assessment of clonal fidelity has been reported in many other plants species such as almond [46], *Swertia chirayita* [47], *Psidium gujava* [48], *Saccharum officinarum* [49], *Gerbera* [50], and *Simmondsia chinensis* [51] using ISSR, RAPD, and SSR markers. In our experiment, the micro-propagated plants of finger millet showed 100% genetic fidelity and similarity with the mother plants using ISSR markers with no somaclonal variation, and this was the first study showing the confirmation using ISSR markers.

## 5. Conclusions

Our study was the first to report the rapid multiplication of shoots (17) using TDZ 4.5 µM for the in vitro propagation of finger millet genotype IE-2606. The study of the results is more efficient, as we were able to obtain more than 15 regenerated plants from a single shoot apex. Obtaining a large number of shoots from a single tissue through the regeneration protocol is critical for transformation experiments. For instance, the GUS assay and PCR analysis require a large number of shoots to verify the transformation status of plants. Furthermore, having a large number of shoots during transgenic experiments allows us to collect more physiological and biochemical characteristics from the transgenic plants. We believed that this study would fulfill all these requirements. Five ISSR primers confirmed the genetic fidelity of the regenerated shoots. There was no somaclonal difference found between the mother plants and the regenerated plants. This procedure might be helpful for producing comparable clones on a wide scale. In order to create the finger millet genotypes for abiotic and biotic stress, it could help with the ongoing supply of plant materials for biotechnological applications.

## Figures and Tables

**Figure 1 biotech-14-00029-f001:**
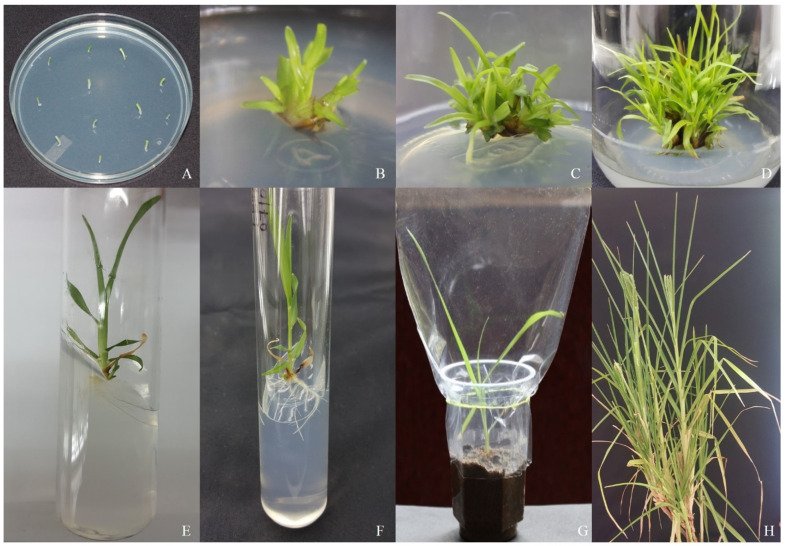
Direct regeneration from shoot apex explants of the finger millet genotype. (**A**) Shoot apex explants in MS media supplemented with B5-vitamins, FeEDTA, and 4.5 µM TDZ for direct regeneration in finger millet genotype IE-2606; (**B**) multiple shoot initiation after 10–12 days; (**C**) rapid multiplication of shoots after 2 weeks of sub-culture; (**D**) maturation and the maximum number of 17 shoots seen after 2 weeks of sub-culture in the same media; (**E**) elongated shoots; (**F**) root development in half-strength MS medium containing 2.8 μM IBA after 2 weeks of incubation; (**G**) acclimatized plantlets in plastic cups containing a 1:1 ratio of sterile soil and vermicompost; (**H**) maturation and flower setting with an ear of genotype IE-2606.

**Figure 2 biotech-14-00029-f002:**
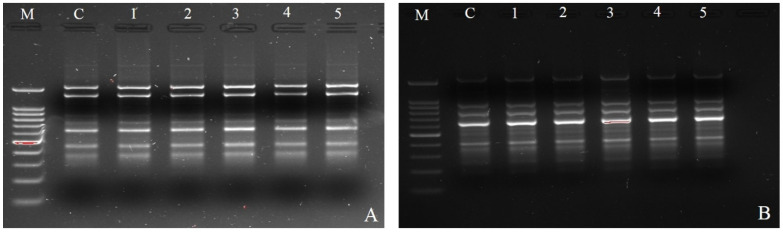
Analysis of mother plant and in vitro regenerated plants obtained from shoot apex explants of finger millet genotype (IE-2606). Amplification product obtained with ISSR primers: (**A**) ISSR-UBC 841, M 100 bp ladder, C mother plant, 1–5 plantlets cultured on MS medium with 6.8 µM TDZ; (**B**) ISSR-UBC 834, M 100 bp ladder, C mother plant, 1–5 plantlets cultured on MS medium with 4.5 µM TDZ.

**Table 1 biotech-14-00029-t001:** Effect of different plant growth regulators on shoot induction from shoot apex explants of finger millet genotype (IE-2606) after 6 weeks of incubation in light.

PGR Used in Shoot Induction Medium (µM)			Name of the Genotype		
BAP	TDZ	Kin	IE-2606		
			Explants Formed Shoots %	Mean Number of Shoots/Shoot Apex Explants	Shoot Length (cm)
2.2	-	-	20.0 ± 4.7 ^d^	3.6 ± 0.2 ^f^	1.6 ± 0.2 ^e^
4.4	-	-	56.6 ± 7.2 ^c^	6.3 ± 0.5 ^e,f^	4.1 ± 0.3 ^c,d^
6.6	-	-	63.3 ± 7.2 ^b,c^	8.0 ± 0.4 ^d,e,f^	5.2 ± 0.2 ^b,c^
8.8	-	-	73.3 ± 2.7 ^a,b,c^	9.3 ± 0.5 ^c,d,e^	5.9 ± 0.3 ^a,b,c^
8.8	-	1.1	26.6 ± 2.7 ^d^	8.3 ± 1.1 ^d,e,f^	1.5 ± 0.1 ^e^
8.8	-	2.3	56.6 ± 2.7 ^c^	12.0 ± 0.9 ^a,b,c,d^	4.3 ± 0.5 ^b,c,d^
8.8	-	3.4	70.0 ± 4.7 ^a,b,c^	14.6 ± 1.1 ^a,b,c^	5.7 ± 0.3 ^a,b,c^
8.8	-	4.6	66.6 ± 2.7 ^b,c^	12.3 ± 1.5 ^a,b,c,d^	4.9 ± 0.3 ^b,c,d^
-	2.2	-	60.0 ± 4.7 ^b,c^	10.0 ± 0. 4 ^b,c,e^	3.1 ± 0.4 ^d,e^
-	4.5	-	96.6 ± 2.7 ^a^	17.3 ± 0.7 ^a^	7.4 ± 0.2 ^a^
-	6.8	-	86.6 ± 2.7 ^a,b^	15.0 ± 0.4 ^a,b,d^	6.3 ± 0.3 ^a,b^
-	9.0	-	76.6 ± 2.7 ^a,b,c^	13.3 ± 1.0 ^a,b,c,d^	6.0 ± 0.1 ^a,b,c^

Different alphabetical letters (a–f) indicate significant differences among the treatment means analysed using SPSS (*p* < 0.05). Values carrying the same alphabet did not vary significantly (*p* ˃ 0.05).

**Table 2 biotech-14-00029-t002:** Effect of IAA and IBA on root induction from regenerated shoots of finger millet genotype IE-2606 on half-strength MS medium, after 2 weeks of incubation in light.

PGRs Used in Root Induction Medium	Concentrations of Auxin (µM)	Rooted Cultures (%)	Mean Number of Roots	Root Length (in cm)
	0.0	5.5 ± 4.5 ^d^	0.3 ± 0.27 ^d^	0.2 ± 0.1 ^d^
IAA	0.25 µM	33.2 ± 7.8 ^c,d^	2.0 ± 0.47 ^c,d^	3.9 ± 0.4 ^c^
	2.85 µM	77.4 ± 4.5 ^a,b^	4.6 ± 0.47 ^a^	6.9 ± 0.4 ^a,b^
	5.70 µM	66.4 ± 7.8 ^a,b,c^	4.0 ± 0.27 ^a,b^	5.6 ± 0.3 ^b,c^
	0.0	5.5 ± 4.5 ^d^	0.3 ± 0.2 ^d^	0.3 ± 0.2 ^d^
	0.25 µM	44.2 ± 4.5 ^b,c^	2.6 ± 0.2 ^b,c^	4.6 ± 0.4 ^c^
IBA	2.46 µM	94.3 ± 4.6 ^a^	5.6 ± 0.2 ^a^	8.2 ± 0.3 ^a^
	4.9 µM	71.9 ± 4.5 ^a,b^	4.3 ± 0.2 ^a,b^	5.7 ± 0.3 ^b,c^

Different alphabetical letters (a–d) indicate significant differences among the treatment means analysed using SPSS. Values carrying the same alphabet did not vary significantly (*p* ˃ 0.05).

**Table 3 biotech-14-00029-t003:** List of different ISSR primers used for detecting clonal fidelity in micro-propagated plants of finger millet.

S. No	Primers Name	Primers Sequences (5′-3′)	Annealing Tm (°C)	Number of Scoreable Bands Per Primer	Monomorphism (%)
1	UBC 834	AGAGAGAGAGAGAGAGYT	50.5	5	100
2	UBC 841	GAGAGAGAGAGAGAGAGC	51.5	7	100
3	UBC 848	CACACACACACACACARG	55.0	5	100
4	UBC 856	ACACACACACACACACYA	55.0	4	100
5	UBC 857	ACACACACACACACACYG	54.0	6	100
**Total number of bands produced**				**27**	

## Data Availability

The original contributions presented in this study are included in the article. Further inquiries can be directed to the corresponding author(s).

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
