# Peer review of "Direct Shoot Regeneration from the Finger Millet’s In Vitro-Derived Shoot Apex and Genetic Fidelity Study with ISSR Markers"

_biotech, 2025, doi:10.3390/biotech14020029_

Round 1

Reviewer 1 Report

Comments and Suggestions for Authors

Article ‘Direct shoot …… ISSR markers’ deals with the optimization of an efficient regenerated protocol for finger millet using shoot apex as an explant. My observations are as follows:–

  1. Different methods/protocol are available for shoot regeneration in finger millets through both direct and indirect organogenesis using different types of explants. What new vale is added by this paper?
  2. Line no. 92: 90 £ 15 mm means?
  3. TDZ, BAP and Kin all are cytokinin and used for shoot regeneration. Authors used BAB alone and also with Kin. Why they not observed cumulative effect of Kin alone, TDZ+Kin, BAP+TDZ+Kin?
  4. Only IAA and IBA used for rooting, why other auxins and different combinations were not tested?
  5. What is basis of selection of ISSR markers?   
  6. Why only Five ISSR markers?
  7. For soma clonal variation or clonal fidelity study, ISSR was used, why not other more reliable markers (e.g. AFLP, RAPD, SCoT or SSR) or technique used?

Author Response

General Comments: Article ‘Direct shoot …… ISSR markers’ deals with the optimization of an efficient regenerated protocol for finger millet using shoot apex as an explant. My observations are as follows: –

Response. Thank you for your suggestions to improve the contents of the manuscript. We have revised the manuscript completely according to your suggestions and properly responded answers for each questions.

Q1. Different methods/protocol are available for shoot regeneration in finger millets through both direct and indirect organogenesis using different types of explants. What new vale is added by this paper?

Response. Thank you for raising the valuable question. We have agreed that there are various methods/protocols for shoot regeneration in finger millet through direct and indirect organogenesis. However, there are more than 5000 finger millet genotypes in the world. Therefore, the tissue culture protocols differ among the genotypes. To date, no one has reported the tissue culture protocol for IE-2606. Therefore, the novelty of the paper is that we are the first to standardize the direct regeneration protocol for finger millet genotype (IE-2606). In addition, compared to other reports, when increasing the time and changing the hormones, our study provides more multiple shoots. Therefore, the protocol developed from this study will help other researchers in their future studies.

Q2. Line no. 92: 90 £ 15 mm means?

Response: It means the size of the petri dish.

Q3. TDZ, BAP and Kin all are cytokinin and used for shoot regeneration. Authors used BAB alone and also with Kin. Why they not observed cumulative effect of Kin alone, TDZ+Kin, BAP+TDZ+Kin?

Response. Previously, Sathish et al. (2015) has already found the effects of multiple shoot induction of BAP+ TDZ which is why did not use the combination in our study. However, there are no reports related to three combinations of cytokinins like BAP+TDZ+Kin. We would like to thank the reviewer for providing this valuable suggestion. We will consider this suggestion and do future work.

Q4. Only IAA and IBA used for rooting, why other auxins and different combinations were not tested?

Response. In general, many researchers have used IAA and IBA to stimulate roots in plants, which is why we used it in our study and obtained some positive results.

Q5. What is basis of selection of ISSR markers?   

Response. In tissue culture raised plants, a number of molecular markers such as restriction fragment length polymorphism (RFLP), simple sequence repeat (SSR), random amplified polymorphic DNA (RAPD), amplified fragment length polymorphism (AFLP) and ISSR have been used to evaluate the clonal fidelity in plant science field. Among these markers, ISSR markers are suitable to identified variations among tissue culture developed plants than other DNA markers include single nucleotide polymorphisms (SNPs). Because, SNPs markers are very costly markers requiring extensive investment in equipment and manpower. So in the present study we have chosen ISSR markers instead of SNPs because of their applications, simplicity and cost-effectiveness.

Q6. Why only Five ISSR markers?

Response. When we started this experiment, there was no genome sequence for finger millet at that time. Apart from this, we felt that it was sufficient to confirm clonal integrity using five ISSR markers. However, we will try to use more markers as per the reviewer's suggestions in our future studies.

Q7. For soma clonal variation or clonal fidelity study, ISSR was used, why not other more reliable markers (e.g. AFLP, RAPD, SCoT or SSR) or technique used?

Response. We agreed with the reviewer suggestions. In tissue culture raised plants, a number of molecular markers such as restriction fragment length polymorphism (RFLP), simple sequence repeat (SSR), random amplified polymorphic DNA (RAPD), amplified fragment length polymorphism (AFLP) and ISSR have been used to evaluate the clonal fidelity in plant science field. Among these markers, ISSR markers are suitable to identified variations among tissue culture developed plants than other DNA markers include single nucleotide polymorphisms (SNPs). Because, SNPs markers are very costly markers requiring extensive investment in equipment and manpower. So in the present study we have chosen ISSR markers instead of SNPs because of their applications, simplicity and cost-effectiveness.

Reviewer 2 Report

Comments and Suggestions for Authors

Dear Authors,

Thank you for the opportunity to review the manuscript titled "Direct shoot regeneration from the finger millet's in vitro-derived shoot apex and genetic fidelity study with ISSR markers"

The paper is well-structured, and the topic addressed is relevant to the field.

Some remarks:

Keywords:

Some keywords already appear in the title. I recommend either removing these keywords or replacing them with a more specific or related term that adds value and improves discoverability.

  1. Introduction

The introduction provides a comprehensive overview of the importance of finger millet from both nutritional and agronomic perspectives. The rationale for employing in vitro techniques is well presented, and the discussion of previous studies helps to establish the research gap.

I recommend revising the phrase “The ultimate goal of the work...” as the current wording may suggest ambiguity about the objective. Phrases such as “The aim of this study...” or “This study aims to...” are more precise and help maintain focus when presenting research objectives.

  1. Materials and Methods

This section is generally well structured, but there are a few aspects that could be improved:

Please indicate the type of culture vessels, the volume of culture medium per vessel, and the number of explants per culture vessel in sections 2.2. Propagation and elongation of shoots and 2.3. Root induction.

It would be useful to specify what criteria were used to select the 5 regenerated plants (random? representative?) in section 2.5. Analysis of ISSR primers.

Specify whether the values in the text and tables are means ± standard error (SE) or standard deviation (SD) in section 2.6. Statistical analysis

  1. Results

In section 3.6. Examining the clonal fidelity of in-vitro regenerated plants by ISSR markers you refer to Fig 2A and 2B but this figure does not appear in the text. Please clarify.

I also recommend that you insert an image of a representative gel in this section

  1. Discussion

There are several Latin names in this section that you should italicize: line 284-286: line 294, line 305-306.

Notes for the entire text:

Please check throughout the text for uniformity of units of measurement, namely the space between quantity and unit of measurement (ex: Maximum number of shoot induction (17.3) was observed in 4.5 µM of TDZ followed by 2.2 µM (10.0), 6.8 µM (15.0) and 9.0 µM (13.0) of TDZ not .... 2.2µM, 6.8µM and 9.0µM.

Also, use uniform writing when using the ± and +symbols, either with spaces (8.8 + 3.4 µM) or without spaces (8.8+1.1µM BAP).

Author Response

General Comments: Thank you for the opportunity to review the manuscript titled "Direct shoot regeneration from the finger millet's in vitro-derived shoot apex and genetic fidelity study with ISSR markers" The paper is well-structured, and the topic addressed is relevant to the field.

Response. Thank you for your suggestions and providing positive feedback on our manuscript. We have improved the contents of the manuscript as per your suggestions and responded all your comments properly.

Q1. Some keywords already appear in the title. I recommend either removing these keywords or replacing them with a more specific or related term that adds value and improves discoverability.

Response. Thank you for your suggestion. We have removed two keywords those are already appear in the title.

Q2. The introduction provides a comprehensive overview of the importance of finger millet from both nutritional and agronomic perspectives. The rationale for employing in vitro techniques is well presented, and the discussion of previous studies helps to establish the research gap. I recommend revising the phrase “The ultimate goal of the work...” as the current wording may suggest ambiguity about the objective. Phrases such as “The aim of this study...” or “This study aims to...” are more precise and help maintain focus when presenting research objectives.

Response. Thank you for your valuable suggestion. We have rephrased “The ultimate goal of the work” into “The aim of the study” as per your suggestion in the introduction section.

Q3. This section is generally well structured, but there are a few aspects that could be improved: Please indicate the type of culture vessels, the volume of culture medium per vessel, and the number of explants per culture vessel in sections 2.2. Propagation and elongation of shoots and 2.3. Root induction.

Response. Thank you for your suggestions. We have included all the necessary information as per your suggestions.

“Glass bottles were used as a culture vessel to maintain the plants and about 30-50 ml of culture was maintained in each bottle to maintain the plant culture. Initially, about 3 explants at each concentration were maintained in each bottle for multiple shoot induction. After multiple shoots were initiated, they were maintained in a single bottle for multiple shoot induction. For induction of roots, plants were maintained in 100 ml test tubes. Single plant was maintained in each tube”.  

Q4. It would be useful to specify what criteria were used to select the 5 regenerated plants (random? representative?) in section 2.5. Analysis of ISSR primers.

Response. As per your suggestion, we have included the necessary details in the section 2.5.  

Q5. Specify whether the values in the text and tables are means ± standard error (SE) or standard deviation (SD) in section 2.6. Statistical analysis

Response. Thank you for your comments. We have included as per your suggestion.

Q6. In section 3.6. Examining the clonal fidelity of in-vitro regenerated plants by ISSR markers you refer to Fig 2A and 2B but this figure does not appear in the text. Please clarify. I also recommend that you insert an image of a representative gel in this section

Response. Sorry for that. We forgot to include in the previous version. Now we have included for publication.

Q7. There are several Latin names in this section that you should italicize: line 284-286: line 294, line 305-306.

Response. Thank you for your suggestions. We have italicized the plants name as per your suggestion

Q8. Please check throughout the text for uniformity of units of measurement, namely the space between quantity and unit of measurement (ex: Maximum number of shoot induction (17.3) was observed in 4.5 µM of TDZ followed by 2.2 µM (10.0), 6.8 µM (15.0) and 9.0 µM (13.0) of TDZ not .... 2.2µM, 6.8µM and 9.0µM. Also, use uniform writing when using the ± and +symbols, either with spaces (8.8 + 3.4 µM) or without spaces (8.8+1.1µM BAP).

Response. Thank you for evaluating our manuscript in-depth way. We have rectified the typo issues throughout the manuscript as per your suggestions.

Round 2

Reviewer 1 Report

Comments and Suggestions for Authors

Revision is satisfactory.